# Analysis of Impulse Control Disorders (ICDs) and Factors Associated with Their Development in a Parkinson’s Disease Population

**DOI:** 10.3390/healthcare9101263

**Published:** 2021-09-24

**Authors:** Mauricio Iván García-Rubio, María Elisa Otero-Cerdeira, Christian Gabriel Toledo-Lozano, Sofía Lizeth Alcaraz-Estrada, Juan Antonio Suárez-Cuenca, Ramón Mauricio Coral-Vázquez, Paul Mondragón-Terán, Juan Antonio Pineda-Juárez, Luis Fernando Díaz-López, Silvia García

**Affiliations:** 1Movement Disorders & Neuroscience Unit, (UTMON) Hospital Español, Ciudad de México 11520, Mexico; mauricio.garcia@utmon.com.mx (M.I.G.-R.); neurotm@utmon.com.mx (M.E.O.-C.); 2Clinical Research Department, Centro Médico Nacional “20 de Noviembre”, Ciudad de México 03229, Mexico; drchristiantoledo@gmail.com (C.G.T.-L.); suarej05@gmail.com (J.A.S.-C.); luislong_1805@hotmail.com (L.F.D.-L.); 3Genomic Medicine Division, Centro Médico Nacional “20 de Noviembre”, Ciudad de México 03229, Mexico; sofializeth@gmail.com; 4Sección de Estudios de Posgrado e Investigación, Escuela Superior de Medicina, Instituto Politécnico Nacional, Ciudad de Mexico 11340, Mexico; rcoral@ipn.mx; 5Research Coordination Centro Médico Nacional “20 de Noviembre”, ISSSTE México City, Ciudad de México 03229, Mexico; paul.mondragon@issste.gob.mx (P.M.-T.); juan.pineda@issste.gob.mx (J.A.P.-J.)

**Keywords:** Parkinson’s disease, impulse control disorders, associated factors

## Abstract

Parkinson’s Disease (PD) is a neurodegenerative disease in which non-motor symptoms may appear before motor phenomena, which include Impulse Control Disorders (ICDs). The objective of this study is to identify factors associated with the development of ICDs in PD. An analytical, cross-sectional study was conducted using clinical records from patients diagnosed with PD, both genders, from 40 to 80 years old. Clinical and demographic data were collected: 181 patients were recruited; 80 of them showed PD and ICDs, and they constituted the study group, whereas 101 patients with PD without ICDs constituted the control reference group. The duration of PD was longer in the group with ICDs (*p* < 0.008), and all patients showed at least one ICD: binge eating (61.29%), compulsive shopping (48.75%), hypersexuality (23.75%), gambling behavior (8.75%), and punding (3.75%). After logistic regression analysis, only the use of dopamine agonists remained associated with ICDs (*p* < 0.001), and the tremorgenic form was suggested to be a protective factor (*p* < 0.001). Positive associations were observed between the rigid-akinetic form and compulsive shopping (*p* < 0.007), between male and hypersexuality (*p* < 0.018), and between dopamine agonists and compulsive shopping (*p* < 0.004), and negative associations were observed between motor fluctuations and compulsive shopping (*p* < 0.031), between Deep Brain Stimulation and binge eating (*p* < 0.046), and between levodopa consumption and binge eating (*p* < 0.045). Binge eating, compulsive shopping, and hypersexuality were the most frequent ICDs. Complex forms and motor complications of PD were associated with the development of ICDs.

## 1. Introduction

Parkinson’s disease (PD) is the second most frequent neurodegenerative process, only after Alzheimer’s disease [1] and affects more than 5 million people around the world [2]. Its etiology has been attributed to genetic and environmental factors. The pathogenic role of alpha-synuclein protein plays a key role in the current understanding of PD [3,4]. PD affects all populations, though it is more common in Caucasian males in Europe and North America, while African Americans and other races are less likely to have PD [2,5]. The prevalence of PD in the general population is 0.3% and may affect 1–2% of patients older than 60 years old and up to 4% of patients older than 85, while this prevalence is expected to double by 2030 [6].

Motor symptoms occur after 5 to 10 years from the onset of dopaminergic neuronal loss, and they are still the milestone of PD diagnostic criteria, with bradykinesia, rigidity, tremor, and loss of postural reflexes being clinical manifestations that sustained PD diagnosis [7]. Consequently, the late appearance of motor symptoms delays clinical diagnosis [8].

Non-Motor Symptoms of PD (NMSPD) appear years before motor phenomena and could be the key to early diagnosis. NMSPD are diverse and range from sensory phenomena (hyposmia and anosmia) to behavior disorders, among others. NMSPD include neuropsychiatric symptoms such as depressive disorders (4 to 70%), cognitive decline (32%), excessive daytime drowsiness (51%), Rapid Eye Movement sleep disturbances (27 to 32%), and hallucinations and visual illusions (40%) [9,10,11,12,13,14]. More than 60% of PD patients have one or more psychiatric symptoms, which can precede the onset of motor symptoms, even by years. Some symptoms, such as depression or anxiety, are independent of treatment, while others, such as psychosis and Impulse Control Disorders (ICDs), are triggered by dopamine therapy [15,16].

One group of neuropsychiatric symptoms include ICDs, which are characterized by dysfunctions, in both emotional and behavioral regulation, that share the inability to resist an urge or temptation to carry out actions that result as being harmful for themselves or others [17,18]. Understanding risk factors and how they link to ICDs in PD could allow for the early identification of populations at risk and/or to design therapies leading to better care and to increased quality of life.

Potential risk factors for the onset of ICDs in PD are personality traits or impulsivity, depression, male gender, substance abuse, younger age, an earlier PD onset age, concurrent use of levodopa and dopamine agonists (DAAs), longer duration of DA treatment, being single, history of ICDs before PD, current smoking, more formal education, family history of substance abuse or ICDs, preserved executive functions, increased aggressiveness, irritability, disinhibition, and eating disorders [15,19].

It is not currently clear whether certain ethnicities are more prone to ICDs in PD, but there could be differences in the most common impulsive behaviors of different regions that are culturally driven. Previous studies have found PD-related ICDs in various populations: 60% reported hypersexuality (Brazil, Finland, and Israel), >50% described punding (Turkey, Mexico, and India), 57% reported compulsive shopping (Brazil), and ~40% reported gambling (Italy and the USA) [20].

PD-related ICDs have been associated with stereotyped behavior or punding in 13–26% of the cases [14]. Both manifestations are closely linked to agonist dopamine drug use [21,22,23]. Within the ICDs, gaming disorders and hypersexuality are more common in males, while binge eating and compulsive shopping are more common in women [24,25]. ICDs in any of the subtypes are usually extremely devastating to the patient and their family’s quality of life. A summary of the factors in the PD profile is shown in Figure 1.

DAAs (pramipexole, rotigotine, and ropinirole) are an important group of drugs for PD treatment [26,27]. Some authors propose that ICDs may be caused by an impaired capacity of the orbitofrontal cortex (OFC) to guide behavior [28], and that has been attributed to neuronal dopaminergic degeneration, facilitating ICD occurrence in dopamine replacement therapies [29].

In Mexico, PD is a growing health challenge in which there is not enough information about ICD frequency and the factors involved. The goal of this research is to identify factors associated with ICD development in PD patients treated in Mexico City.

## 2. Materials and Methods

An analytical, cross-sectional (case–control) study was conducted using the clinical records of patients from the Movement Disorder & Neuroscience Unit (UTMON) of the “Hospital Español de México” in Mexico City. The ethics committee approved the study.

Patients with PD diagnosis, according to the United Kingdom Parkinson’s Disease Society Brain Bank (UK-PDSBB), were recruited. The sampling was non-probabilistic for convenience. The inclusion criteria included patients of both sex were between 40 and 80 years old, and how did not have a psychiatric history before PD diagnosis.

Patients were divided into two groups: the ICD Group “PD with ICDs”, who were diagnosed by a certificated psychiatrist according to DSM-5 criteria, and the Non-ICD Group, “PD without ICDs”. Patients with dementia were excluded, as were those with non-congruent data in clinical records. Information about both groups was obtained from the same source, which is the Clinical Files Department of “Hospital Español de México”. The cases were selected consecutively, and the controls were matched by age and gender.

Clinical and demographic data regarding the onset and evolution of PD was collected. In both groups, the type of ICD and specific psychiatric symptoms were documented, as were the time of application of antiparkinsonian-type drugs, their dose, and their time of use; the use of Deep Brain Stimulation (DBS); and the UPDRS score.

Association analyses were performed with clinical-demographic data, therapy variables, and ICD presence. A *t*-test was used to compare baseline characteristics. Bonferroni correction was used as a method to adjust p-values due to multiple comparisons in post hoc analysis. Logistic regression analyses were performed and expressed as Odds Ratio and 95% confidence intervals. The Statistical Package for the Social Sciences (SPSS) software v.24 (IBM, Armonk, NY, USA) was used for statistical analysis.

## 3. Results

One hundred and eighty-one patients with PD were recruited. The study group (ICD Group) showed PD and concomitant ICDs (*n* = 80). For comparison, a control reference (Non-ICD Group) was included (*n* = 101).

The demographic data are comparable in both groups (Table 1); however, the PD duration as significantly longer in the ICD Group. Bonferroni correction was used as a method to adjust *p*-values due to multiple comparisons in a post hoc analysis.

According to the outcome ICDs (Table 2), it was remarkable that the tremorgenic type of PD was more common in the Non-ICD Group (*p* < 0.0001); the rigid-akinetic form was more frequent in the ICD Group (*p* < 0.005). As expected, the high prevalence of tremorgenic manifestations was negatively associated with the development of ICDs (*p* < 0.006), while a high prevalence of rigidity positively was associated with ICDs (*p* < 0.005).

Regarding the doses of dopamine replacement therapy used (levodopa *p* < 0.166, pramipexole *p* < 0.232, rotigotine *p* < 0.262, and rasagiline *p* < 0.041), no statistically significant difference was found, except for the doses of rasagiline.

The logistic regression analysis evidenced that only DAAs use was significantly associated with ICD development (*p* < 0.001). PD duration showed a non-significant tendency (*p* < 0.067). The tremorgenic form was suggested as a protective factor (*p* < 0.001). In the same analysis, the main non-motor symptoms (depression (*p* < 1.000), anxiety (*p* < 0.515), chronic pain (*p* < 0.335), REM sleep disturbances (*p* < 1.000), hallucinations (*p* < 1.000), and hyposmia (*p* < 0.620)) were not associated with ICDs in our study population.

Five clinical events were more frequent and were associated with ICDs: motor fluctuations (*p* < 0.031); dyskinesias (*p* < 0.029), DAAs consumption (*p* < 0.0001), Parkinson’s disease duration (years), and DBS placement (*p* < 0.029). Surprisingly, the ICD Group consumed the highest doses of rasagiline. Finally, the Hoehn and Yahr Scale score was not associated with ICDs (Figure 2).

Regarding clinical manifestations, the ICD Group was characterized by the following profile: binge eating was highly prevalent (*n* = 47, 58.8%), followed by compulsive shopping (*n* = 22, 21.4%), hypersexuality (*n* = 19, 18.4%), gambling disorder (*n* = 3, 2.9%), and punding (*n* = 4, 5.1%). According to the number of ICDs, one ICD type was developed in 49 out of 80 cases (61.25%): binge eating disorder (*n* = 22, 27.5%), a compulsive shopping disorder (*n* = 15, 18.75%), hypersexuality (*n* = 8, 10%), gambling disorder (*n* = 2, 2.5%), and punding disorder (*n* = 2, 2.5%). Cases with 2 ICDs occurred in 23 out of 80 cases (28.75%), as follows: binge eating disorder and compulsive shopping disorders (*n* = 12, 15%), hypersexuality and binge eating disorder (*n* = 5, 6.25%), binge eating disorder and gambling disorder (*n* = 2, 2.5%), compulsive shopping disorder and hypersexuality (*n* = 2, 2.5%), and gambling disorder and compulsive shopping disorder (*n* = 2, 2.5%). Only eight patients (10%) developed three ICDs types; six (7.5%) developed binge eating disorder, hypersexuality, and compulsive shopping disorder; one patient (1.25%) developed hypersexuality, compulsive shopping disorder, and punding disorder; and one (1.25%) developed hypersexuality, compulsive shopping, and gambling disorder.

The following variables were positively associated: male and hypersexuality; mixed form and gambling behavior; rigid-akinetic form and compulsive shopping, DAAs and compulsive shopping; DAAs and binge eating; motor fluctuations and compulsive shopping; and DBS and binge eating (Table 3). In contrast, negative associations were found between DAAs and hypersexuality, levodopa consumption and binge eating, and tremorgenic form and compulsive shopping.

After adjusting for potential confounding variables, logistic regression showed independent associations between the rigid-akinetic form and compulsive shopping; between the genders male and female, and hypersexuality; between levodopa consumption and binge eating; between DAAs and compulsive shopping; between DAAs and hypersexuality; between DAAs and binge eating; between motor fluctuations and compulsive shopping; and between DBS and binge eating (Table 4).

Additionally, age at PD diagnosis in the ICD Group with hypersexuality (58.12 ± 14.64 years old) was younger than in the Non-ICD Group (63.55 ± 11.820) (*p* < 0.05). The subgroups with compulsive shopping and binge eating also showed differences (59.28 ± 14.74 vs. 63.85 ± 11.403, *p* < 0.04, and 7.96 ± 6.376 vs. 6.01 ± 5.202, *p* < 0.039, respectively); however, no significant association was further observed.

## 4. Discussion

ICDs are a very complex entity in the PD context and need to be considered a multifactorial alteration encompassing treatment drugs for PD and other disease-related factors [17].

In the Diagnostic and Statistical Manual of Mental Disorders, Fifth Edition (DSM-5), ICDs are called “Disruptive, Impulse Control and Behavior Disorders”; these symptoms have a common denominator of an inability to resist an impulse or temptation to perform an act that is harmful [26,30], experiencing pleasure, gratification, or release of tension after performing the action. ICDs reported in PD patients include compulsive shopping, gambling, hypersexuality, obsessing over hobbies, punding, binge eating, and compulsive medication use [31,32].

NMSPDs such as depression, anxiety, REM sleep disturbances, aggressiveness, chronic pain, or hyposmia have been described as being closely related to the development of ICDs [17]. In the present study population, such NMSPDs did not show particular differences between comparison groups, perhaps because these disorders are common in PD but do not necessarily predict the appearance of ICDs.

One of the main findings of this study was the association between rigid-akinetic, tremorgenic forms with ICDs. Dopamine is involved in both PD and ICDs and is a very important substance in the normal brain-specific function for the control of motor function, motivation, and learning. Although nigrostriatal denervation was proposed as an etiology for ICDs, only a few patients with PD develop them. The nigrostriatal pathway is involved in motor function and influences the mesocorticolimbic pathway (reward system) and the tuberoinfundibular pathway, which influences the secretion of prolactin [26]. The neuro-degenerative processes of PD affect several pathways responsible for modulating the reward system, positive reinforcement, motivation, inhibitory control, and decision making implicated in impulse control [24]. Despite the theoretically limiting actions in certain regions of the Central Nervous System of DAAs, it habitually affects different areas of the brain in PD patients. For more than two decades, ICD cases have been associated with DAAs consumption [33,34] and they were considered iatrogenic, so symptomatology began after using DAAs and disappeared when it was stopped [26]. Elucidating how biological events of the disease interact with the effect of drugs for general ICDs is a challenge. There are researchers who consider that impulsivity is a PD symptom [35,36]. However, DAAs influence hyperactivation of the reward system and reduce neuronal activity related to impulse control processes and response inhibition. For the association found between levodopa and binge eating, previous studies have suggested that reward systems may have a role in binge eating. Nonetheless, a direct relationship between the use of levodopa and this ICD [37] has not been established. The association between DAAs and the development of binge eating has been shown in the literature, mainly with pramipexole, but the exact action mechanism is still unknown [38]. Male gender is an independent factor for developing ICDs [39,40,41,42,43,44] implicated in gambling and hypersexual behavior [45]. In our population, male gender was associated with hypersexuality, but there was no association in females. Other studies have described the presence of hypersexuality in patients treated with levodopa, DAAs, or selegiline [46], but that association was not detected in our study.

Likewise, we found that DBS placement was a risk factor for ICDs as well as the association between DBS and binge eating. Current evidence is scarce and controversial. Some studies have described the association of DBS with the improvement, deterioration, or development of ICDs. However, DBS placement in the subthalamic nucleus has been associated with the development of this ICD subtype [47,48]. Stimulation in this nucleus leads to a decrease in the DAAs dose and translates into a decreased appearance of ICDs [49]. In our study population, 13 out of 19 cases with DBS have ICDs, suggesting that DBS could favor the presence of ICDs, perhaps due to the complex physiology and interactions of nuclei of the base, particularly of the ventral striatum most frequently associated with stimulation of the subthalamic nucleus [49,50].

Another relevant finding was the duration of PD. Previous reports show that younger patients (under 65 years old) with PD or its onset at a younger age is an independent predictor for developing ICDs [24,40,44,51,52,53,54,55,56,57,58]. However, no association of PD duration or age of onset with gambling was detected in the present study. It is possible that more attention is paid to patients with movement, whereas ICDs may be uncovered. Another possibility is that the age of the study population (mean age 60 years old) could constitute a bias selection. Regarding the duration of PD associated with the development of ICDs [35,42,58], there is no general consensus between authors [59]; it is possible that more extended degeneration leads to more tissue damage. Consistently, PD diagnosed at a younger age was related with hypersexuality and compulsive disorders, while PD duration was longer in binge eating cases. It was interesting to find a positive association between higher doses of rasagiline and the development of ICDs. There are reports suggesting that rasagiline is a risk factor for ICDs [53,60,61]. Studies specifically designed to confirm this finding are needed. In contrast, DAAs and L-dopa have been involved in ICDs developed in PD patients [23,54,62,63,64,65,66,67,68], which is consistent with our results.

An important observation in this research was that the rigid-akinetic form was a risk factor for the presence of ICDs, while the tremorgenic form seems to protect against the presence of ICDs, according to association analyses. It has been documented that patients with greater motor disease complexity and more frequent motor fluctuations have ICDs. Furthermore, PD tremors seem to be strongly related to abnormal activity in the cerebellum receptor of the thalamus gland (ventral intermediate nucleus). Tremors respond to dopaminergic therapy, although it is not clear how the loss of dopamine is related with tremors. Dopamine may have less importance in tremors than in other PD symptoms and is a protective factor for the development of ICDs [43,45,69,70]. Based on current scientific evidence, current proposed pathophysiological mechanisms explaining these findings are complex and need future research for a better understanding [49]. A strength of this research is that it included subjects from diverse ethnical origins: American, Arab, Spanish, French, Jewish, Lebanese, and Mexican Mestizo, due to the type of Hospital target population, where Mexican Mestizo was the most common population. There were no significant differences despite ethnical variability, suggesting that ICDs behave similarly between ethnical populations, at least from the ethnical diversity included in the present study.

The limitations of this study are related to the inclusion of patients from a center specialized in the care of PD, which could explain the high prevalence of ICDs in our study and the lack of standardized registries. At the time of this study, there are no validated diagnostic instruments in the Mexican population to identify patients with ICDs. There are no epidemiological or statistical records about ICDs in patients with PD that allow us to establish a comparison with the results of this research in the Mexican population. Furthermore, the authors consider it relevant to replicate the findings in a larger sized sample to establish more precise associations, allowing for comparison with American or European studies. Likewise, it is necessary to carry out prospective studies to generate more solid evidence on the impact of social, cultural, genetic, or environmental factors that influence the development of these disorders in the Mexican population beyond dopamine replacement therapy.

It is important to recognize factors associated with ICDs since they have a significant impact on the quality of life of the patient and their family members. Early detection allows for timely treatment of these disorders to be started and for avoiding consolidation of ICDs.

## 5. Conclusions

In conclusion, the most frequent ICDs observed in this study were binge eating, compulsive shopping, and hypersexuality. The rigid-akinetic form of PD, DAAs use, and motor fluctuations were positively associated with compulsive shopping devolvement; in contrast, the tremorgenic form was a protective factor. Levodopa and DAAs use were associated with binge eating devolvement. DBS placement was a risk factor for general ICDs. In addition, being male and DAAs use were risk factors for hypersexuality devolvement.

## Figures and Tables

**Figure 1 healthcare-09-01263-f001:**
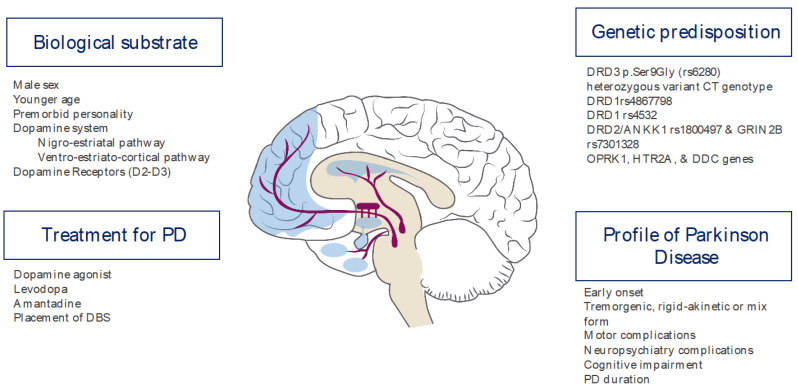
Risk factors for ICDs in patients with PD.

**Figure 2 healthcare-09-01263-f002:**
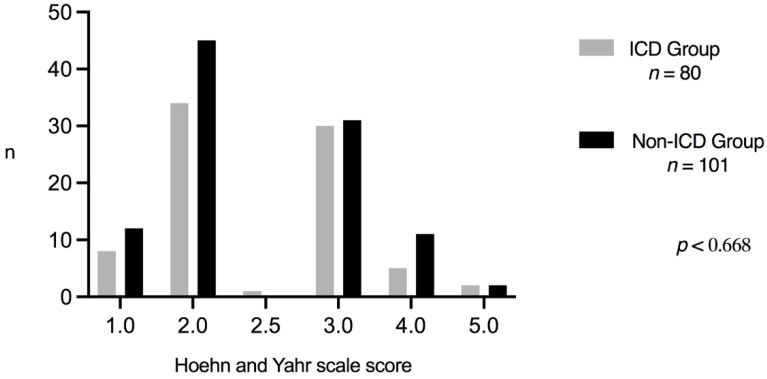
Hoehn and Yahr comparison.

**Table 1 healthcare-09-01263-t001:** Characteristic demography of patients in each group.

Demographic Characteristics	ICD Group (*n* = 80)	Non-ICD Group (*n* = 101)	*p*-Value
Gender, *n* (%)			
Male	48 (60)	55 (54.5)	0.454
Female	32 (40)	46 (45.5)	
Scholarship *n* (%)			
Without schooling	0 (0)	2 (2)	0.313
Elementary School	10 (12.25)	5 (5)	
Secondary school	4 (5)	5 (5)	
High School	7 (8.8)	17 (16.8)	
College	34 (42.5)	43 (42.6)	
Master	5 (6.3)	7 (6.9)	
Doctorate	20 (25)	22 (21.8)	
Employment, *n* (%)			
Retired	22 (27.5)	16 (15.8)	0.323
Housewife	17 (21.3)	19 (18.8)	
Employee	11 (13.8)	20 (19.8)	
Professional	28 (35)	43 (42.6)	
Entrepreneur	2 (2.5)	3 (3)	
Ancestry *n* (%)			
American	1 (1.39)	1 (1)	0.868
Arab	0 (0)	1 (1)	0.372
Spanish	22 (27.5)	19 (18.8)	0.166
French	2 (2.5)	0 (0)	0.110
Jewish	4 (5)	5 (55.6)	0.988
Lebanese	1 (1.3)	0 (0)	0.260
Mestizo Mexican	50 (62.5)	75 (74.3)	0.106
Laterality, *n* (%)			
Right-handed	80 (100)	97 (96)	0.98
Left-handed	0 (0)	3 (3)	
PD duration (years) *	7.75 ± 6.24	5.53 ± 4.793	0.008
Age when PD started (years) *	69.71 ± 12.622	69.12 ± 11.224	0.380

* Mean and standard deviation.

**Table 2 healthcare-09-01263-t002:** Clinical characteristic of both groups and their association with the outcome Impulse Control Disorders (ICDs).

Clinical Characteristics	ICD Group (*n* = 80)	Non-ICD Group (*n* = 101)	*p*-Value	OR	95% CI
PD clinical predominant type, *n* (%)					
Tremorgenic form	23 (28.75)	39 (38.61	0.0001	0.548	0.399–0.803
Rigid-akinetic form	33 (41.3)	21 (20.8)	0.005	1.610	1.188–2.200
Mixed form	6 (7.5)	2 (1.98)	0.141	1.753	1.134–2.711
Symptoms *n* (%):					
Tremor	47 (58.75)	79 (78.21)	0.006	0.622	0.455–0.850
Bradykinesia	78 (97.5)	100 (99)	0.584	0.657	0.290–1.488
Rigid	45 (56.25)	41 (40.59)	0.051	1.420	1.019–1.979
Postural instability	9 (11.25)	13 (12.87)	0.821	0.916	0.539–1.558
Medical history *n* (%)					
Arterial hypertension	57 (71.25)	37 (36.63)	0.181	1.199	0.921–1.561
Mellitus diabetes	13 (1.25	7 (6.93)	0.477	1.189	0.837–1.689
Dyslipidemia	18 (22.5)	8(7.9)	0.200	1.293	0.961–1-737
Hypothyroidism	12 (15)	4 (3.9)	0.121	1.390	1.014–1.907
Dopamine replacement therapy *n* (%)					
Levodopa	74 (92.5)	85 (84.2)	0.110	1.706	0.845–3.445
Dopamine Agonists	72 (90)	54 (53.5)	0.0001	3.929	2.034–7.587
MAOI	53 (66.3)	66 (65.3)	1.000	1.028	0.723–1.461
COMI	9 (11.3)	5 (5)	0.161	1.512	0.985–2.321
Motor fluctuations *n* (%)	37 (46.3)	31 (30.7)	0.031	1.469	1.066–2.023
Dyskinesia *n* (%)	29 (36.3)	21 (20.8)	0.029	1.490	1.083–2-049
DBS *n* (%)	13 (16.3)	6 (5.9)	0.029	1.654	1.159–2.362
Non-motor symptoms *n* (%)	*n* = 17	*n* = 17	*p*-Value	OR	95% CI
Depression	1 (5.8)	1 (5.8)	1.000	1.500	0.315–7.137
Anxiety	2 (11.7)	1 (5.8)	0.515	2.333	0.738–7.381
Chronic pain	2 (11.7)	7 (41.1)	0.335	0.444	0.109–1.811
REM sleep disturbances	3 (17.6)	5 (29.4)	1.000	1.125	0.311–4.071
Hallucinations	1 (5.8)	1 (5.8)	1.000	1.500	3.15–7.137
Hyposmia	2 (11.7)	6 (35.2)	0.620	0.563	0.138–2.293
Mean and standard deviation					
UPDRS	23.20 ± 6.028	23.51 ± 6.307	0.734		
Age at PD Diagnosis	61.96 ± 13.386	63.58 ± 11.385	0.380		
PD duration (years)	7.75 ± 6.249	5.53 ± 4.796	0.008		
Doses (mg/day)					
Levodopa	568.019 ± 383.459	466.361 ± 541.84	0.166		
Pramipexole	1.140 ± 1.098	0.952 ± 1.001	0.232		
Rotigotine	5.90 ± 1.77	5.09 ± 1.375	0.262		
Ropirinol	0.19 ± 1.68	0			
Rasagiline	0.4019 ± 0.510	0.2575 ± 0.428	0.041		

CI: confidence interval; COMI: catechol-O-methyltransferase inhibitors; DA: dopamine agonists; ICD: Impulse Control Disorders; MAOI: monoamine oxidase inhibitors; OR: odds ratio; UPDRS: Unified Parkinson’s Disease Rating Scale.

**Table 3 healthcare-09-01263-t003:** Clinical factors, and their association with a spectrum of ICDs.

Clinical Factors	Gambling *n* = 7	*p*	Punding *n* = 3	*p*	Compulsive Shopping *n* = 39	*p*	Hypersexuality *n* = 23	*p*	Binge Eating *n* = 47	*p*
Gender (M/F)	3/4	0.466	2/1	1	22/17	1	19/4	0.012	28/19	0.733
Mexican Ancestry	3/7	0.134	2/3	0.673	29/39	0.273	16/23	0.582	29/47	0.137
Tremorgenic form	4/7	0.692	1/3	0.270	15/39	0.0001	15/23	1	26/47	0.107
Rigid-akinetic form	1/7	0.675	1/3	1	20/39	0.001	7/23	1	17/47	0.265
Mix form	2/7	0.032	1/3	0.127	3/39	0.372	1/23	1	3/47	0.430
Levodopa consumption	7/7	0.397	3/3	0.676	36/39	0.254	20/23	0.554	46/47	0.008
DAAs	7/7	0.103	3/3	0.335	38/39	0.0001	20/23	0.055	41/47	0.002
MAOIs	5/7	1	2/3	1	28/39	0.565	17/23	0.486	30/47	0.857
Motor Fluctuations	2/7	0.710	1/3	1	22/39	0.010	10/23	0.648	23/47	0.083
Dyskinesias	2/7	1	1/3	1	15/39	0.106	7/23	0.804	18/47	0.087
Postural instability	2/7	0.203	0/3	1	2/39	0.170	4/23	0.491	5/47	0.801
DBS	0/7	1	0/3	1	7/39	0.135	5/23	0.072	9/47	0.049

DAAs: dopamine agonists; DBS: deep brain stimulation; MAOI: monoamine oxidase inhibitors; M: male; F: female.

**Table 4 healthcare-09-01263-t004:** Clinical and demographic factors, and their association with a spectrum of ICDs.

Clinical and Demographic Factors	N	*p* Value	OR	95% CI	Logistic Regression *
Mixed form/gambling disorders	2/7	0.032	8.65	1.972–37.951	-
Rigid-akinetic form/compulsive shopping	20/39	0.001	2.542	1.481–4.363	0.007
Male gender/hypersexuality	9/4	0.012	3.597	1.275–10.150	0.018
Levodopa consumption/binge eating	46/47	0.008	0.745	0.651–0.852	0.045
DAAs/compulsive shopping	38/39	0.0021	16.587	2.336–117.777	0.004
DAAs/hypersexuality	20/23	0.056	0.890	0.806–0.982	0.095
DAAs/binge eating	41/47	0.002	2.983	1.346–6.612	-
Motor fluctuations/compulsive shopping	22/39	0.010	2.101	1.203–3.668	0.031
Tremorgenic form/compulsive shopping	15/39	0.0001	0.326	0.185–0.574	-
DBS/binge eating	38/47	0.049	0.495	0.286–0.858	0.046

CI: confidence interval; DAAs: dopamine agonists; DBS: deep brain stimulation, OR: odds ratio. * Reference group: Non-ICD Group.

## Data Availability

Datasets analyzed or generated during the study can be requested from the authors.

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
