# Peer review of "Analysis of Impulse Control Disorders (ICDs) and Factors Associated with Their Development in a Parkinson’s Disease Population"

_healthcare, 2021, doi:10.3390/healthcare9101263_

Round 1

Reviewer 1 Report

In this study, authors have evaluated the impulse control disorders in a cohort of patients in Mexico City and investigated the factors associated with impulse control disorders in Parkinson's disease. Given that the literature is dominated by studies in White Caucasians, the efforts to increase data on a diverse population through this study is appreciated. I only have few concerns that the authors can consider to improve their paper;

- In the Introduction, please explain prior studies that have reported the factors associated with impulse control disorders in Parkinson’s disease.

- Given that the ethnicity component is a strong feature of this paper, please briefly explain the ethnic/racial differences in Parkinson’s disease symptoms, particularly impulse control disorders, to support the need for your study.

- In the Discussion, please add a brief Limitations paragraph and the clinical implications of your findings; describe how clinical practice or research can benefit from your findings.

- Was the levodopa equivalent daily dose compared across the two groups to investigate the difference based on the cumulative dosage? If not, please add that.   

- I recommend using another term to label the two groups for ease of read instead of “cases” and “controls”, given the “controls” also are people with Parkinson’s disease.

- Were p values corrected for multiple comparisons? Given that multiple comparisons were performed without a priori hypotheses, a correction may be needed.

- Table 3 can be presented as a Figure to include an illustration in the paper.

- Please make the “p”s italic.

- Page 4, Line 117 it should be “ICD” not “CDI”.

- It should be “dopamine agonists”, not “dopamimetic agonists”.

Reviewer 2 Report

Garcia-Rubio and collaborators presents a work aimed to explore factors associated with ICDs in PD. The topic is of interest since identifying predictors for the development of ICDs is an important unmet need. However, there I have several concerns with too many aspectos of the current version of the work and thus, I cannot consider it in the current form.

Language is terrible. There is a need for an important work with a native english-spesking. But in addition to grammar, the structure and organization make really difficult to follow the text. As few examples:

  1. the sentence refering to alfasynuclein must stard with “ the involvement”. Then, the following sentence is “out of place” because it sounds like they are talking about the prevalence of alfasynuclein involvement….so beyond grammar etc, please try to structure the whole text in a more organized manner.
  2. this sentence makes no sense in english “A remarkable feature in PD are the Impulse Control Disorders (ICDs) which are heterogeneous that include biological substrates for the treatment for PD”. There are too many sentences like this. I strongly encourage to review again and again a manuscript before submiting it to peer review.

I do not understand why in the introduction so much space is devoted to to generalities of PD and, on the contrary, not on ICDs

The definition, examples and conceptualization of ICDs is too superficial and even wrong in some places

An important methodological issue here is that authors just performed a direct comparison between two PD groups (with and without ICDs). This is not the way to approach the central question of this work and on the contrary, it must be done using a logistic regression analysis. Groups comparisons may be useful to explore main differences in some general clinical/sociodemographic variables.

Because it seems that they have a really good dataset with a lot of clinical data I strongly recomend to rebuild this manuscript and work and to go to two main aspects: 1. Description of the phenomenology of ICDs in this cohort and then, explore (using a regression analysis) the variables that better differentistes both groups. When doing it, please take into accoont other variables (if possible) like cognitive status, depressive symptoms, apathy,…

Round 2

Reviewer 2 Report

I have looked over the revised text and consider that it is still a very poor text with multiple errors. That is why I maintain my previous decision to consider this job as reject.

Author Response

We thank you for your kind support in the revision of the manuscript.